# Title: Caring for Children with Cerebral Palsy: A Challenge to Caregivers in Rural Areas of South Africa

**DOI:** 10.3390/children10030440

**Published:** 2023-02-24

**Authors:** Duppy Manyuma, Mary Maluleke, Ndidzulafhi Selina Raliphaswa, Thingahangwi Cecilia Masutha, Mphedziseni Esther Rangwaneni, Takalani Eldah Thabathi, Ndivhaleni Robert Lavhelani

**Affiliations:** Department of Advanced Nursing Science, University of Venda, Private Bag X 5050, Thohoyandou 0950, South Africa

**Keywords:** caregivers, caring, cerebral palsy, challenge, children, rural areas

## Abstract

Background: In South Africa, children with cerebral palsy are nested within a family setting by immediate relatives, particularly in the rural areas. These immediate relatives are regarded as caregivers and are not trained with regard to providing care to children with special needs. Therefore, they have to find ways to adapt to their new roles of caregiving using the available resources. The aim of this paper is to present the challenges encountered by caregivers of children with cerebral palsy in rural areas of South Africa. Methods: This qualitative, explorative, descriptive, and contextual study included 10 caregivers for children with cerebral palsy who were purposively recruited from the three respective hospitals in Vhembe District within Limpopo Province. Data were collected through individual in-depth interviews and analysed using a thematic approach. Ethical considerations and measures to ensure trustworthiness were upheld throughout the study. Results: Four themes emerged from the study, namely economic problems facing caregivers; health problems encountered by caregivers; maltreatment of caregivers by community members, as well as access to transport for caregivers and children. Conclusion: Participants narrated that caring for a child with cerebral palsy is not an easy task for the caregiver, particularly in the rural areas of South Africa. They encounter various challenges as they strive to provide quality care to the children that they are caring for. An investigation is recommended concerning the kind of model which should be developed to support caregivers in caring for children with cerebral palsy in the rural areas.

## 1. Introduction

Cerebral palsy (CP) is a developmental disability commonly found amongst children. It affects their motor functioning, which in most cases results in damaged motor behavior and oral dysfunction [1]. Since CP affects many parts of the human body, people living with this condition need someone to help them with day-to-day activities, such as bathing, feeding, using the toilet, and moving from one place to another.

Furthermore, [1] reports that children with cerebral palsy demand more attention than children without any special need. Majority of caregivers devote their time in providing care to the children and have less time to do things which make them happy, like going out to parties with other women and attending functions. Apart from having time to themselves, [2] stated that the time given to those children without special needs in a household where there is a child with cerebral palsy is minimal, and other children feel unwanted and often develop anger towards their parents. [3] Caregivers assist children with the execution of daily activities, such as bathing, eating, moving, and going to the toilet.

Ref. [4] argues that continuous lifting of children with cerebral palsy often leads to postural imbalances and result in low back pain and various musculoskeletal illnesses amongst the caregivers. Ref. [5] stated that providing care to a child who is living with a cerebral palsy can be traumatic and affect the mental health of caregivers, resulting in various mental health issues, such as stress, anxiety, and depression. Environmental stressors, such as poor access to transportation system and an unsupportive work environment, increase the burden of care for the caregivers of children with cerebral palsy [6]. Ref. [7] found that taking children with cerebral palsy to the treatment centers increases the financial burden that they already experience. The study further revealed that more caregivers support the idea of having community-based workers who can conduct treatment in their homes to reduce the financial burden of travel. Lack of support from family and society leads to unstable and sensitive emotional states for caregivers of children with cerebral palsy [8].

According to [9], providing care to a person who is living with a disability is a complicated and demanding task, which needs one to be fully focused and thus reduces the caregivers’ opportunity of permanent employment [10]. They also believe that caregivers who assume the role of caring at a younger age will in most cases not finish school and their chances of getting well-paid jobs can be compromised. On the other hand, [6] found that matrimonial complications can result from the changes in the roles that partners have to assume when providing care to the new child in the family who is living with a disability. In South Africa, children with cerebral palsy are nursed within a family setting by immediate relatives, particularly in the rural areas [6]. These immediate relatives are regarded as caregivers and are not trained with regard to providing care to children with special needs. Therefore, they have to find ways to adapt to their new roles of caregiving using the available resources [11].

In December 2016, the main author was invited to a year-end function which was organized for the caregivers of CP and observed that all caregivers brought along the children with CP, the reason being that there is no-one to look after them since children are cared for at their homes, due to the lack of centers that can accommodate them within Vhembe District. Despite multiple research projects conducted on the experiences of caregivers of children with cerebral palsy, there is no sufficient information pertaining to the experiences of caregivers in the Limpopo Province, particularly in the rural areas of Vhembe District. Therefore, a qualitative study was conducted to describe the challenges faced by caregivers of children with cerebral palsy in the rural areas of Vhembe district in Limpopo Province.

## 2. Materials and Methods

A qualitative approach using explorative, descriptive, and contextual designs was adopted as described by [12,13]. The qualitative approach was relevant to this paper since its aim was to present the challenges of caregivers who are caring for children with cerebral palsy in the rural areas of South Africa, which could not be quantified. The use of this approach gave the caregivers and opportunity to narrate their challenges without limitation. It also allowed the researcher to probe more from the caregivers, based on the responses that they were providing.

### 2.1. Explorative Design

An exploratory design was relevant as it is “mostly conducted when little information is available about the phenomenon and gives the researcher a chance to explore more” [14]. The challenges that caregivers of children with cerebral palsy encounter were explored and a greater understanding was gained as the caregivers were narrating how caring for a child with CP is a challenge to them.

### 2.2. Descriptive Design

Descriptive studies aim at describing how the phenomena under investigation are related to one another [15]. The caregivers were given enough time to describe the challenges that they encounter on a daily basis, as a result of providing care to the children who are living with CP.

### 2.3. Contextual Design

The study focused only on those caregivers who are caring for children with cerebral palsy in Vhembe District. Other forms of disabilities and issues which were not aligned to the purpose of the study were not entertained throughout the study. The study was contextual since interviews were conducted at the homes of the caregivers who are caring for children with cerebral palsy, where the actual caring takes place.

### 2.4. Setting

The study was conducted in Vhembe District of Limpopo Province. Vhembe District is situated in the northern part of Limpopo Province. It is regarded as an environment that is primarily rural. There are three centers which cater for children with a disability around the district, and they operate as day care centers. A small number of children are admitted due to the limited space at the centers, forcing most the children to remain in the care of their parents and/or relatives.

### 2.5. Recruitment and Sampling of Participants

After obtaining permission to conduct the study from the Chief Executive Officers, occupational therapists who are responsible for facilitating cerebral palsy support groups from the three selected hospitals were requested to assist in arranging a meeting with the caregivers. During the meetings, caregivers were informed of the study purpose. It was clearly explained to the caregivers that not all of them will be included in the study, as there is a certain number which is desired for the study. The inclusion criteria were explained to all the caregivers who were present. Those who met the criteria of inclusion in the study and agreed to participate were subjected to further selection by the researchers, considering the following factors: the person is above the age of 18 years, residing within Vhembe District, can communicate in English or Tshivenda, and has been caring for a child with cerebral palsy for more than six months; the caregiver attends CP support groups offered by the Occupational Therapy Department in any of the selected hospitals; the caregiver should be caring for a child who cannot move on their own or do any other activity without the assistance of the caregiver. The exclusion criteria applied in this study stipulated that caregivers who were not willing to participate were excluded.

### 2.6. Ethical Considerations

Ethical clearance was obtained from the University of Venda ethics committee (SHS/18/PH/05/0405), Limpopo Provincial Department of Health, Vhembe District Department of Health, and the three selected hospitals issued letters to allow the researchers to recruit participants from their institutions. All participants were provided with the information sheets and consent forms before they participated in the study. Researchers ensured that participants were not physically, emotionally, or psychologically harmed during their participation in the study by avoiding use of insensitive language. Codes were used instead of the participants’ real names in order to protect their identity. Study information was kept away from those who are not part of the study.

### 2.7. Data Collection

Participants were visited several times at their homes prior to data collection, in order to build trust as a way of ensuring prolonged engagement. Individual in-depth interviews were conducted with the caregivers of children with cerebral palsy, who were purposively sampled by the researchers. The study was guided by one central question, “Kindly share the challenges that you encounter on a daily basis as you are caring for a child with cerebral palsy”. This was followed by probing questions depending on the participant’s response. Interviews were conducted at the home of the participants. During interviews, various communication skills, such as probing, paraphrasing, and summarizing, were used to understand the challenges that caregivers of children with cerebral palsy encounter in the rural areas of Vhembe District in South Africa. At the end of each interview, the audio-recorder was played back for the participants to verify the recorded information and to identify the gaps, as a way of ensuring a member check. Data collected in Tshivenda were translated into English and transcribed verbatim. Interviews were conducted individually with participants until no new information was coming in. Data saturation was reached with the eighth participant. However, two more participants were added to contribute supplementary information during the interviews. Each interview lasted for approximately 45–50 min.

### 2.8. Data Analysis

The thematic approach by Creswell [15] was used to analyze data gathered from the caregivers through individual in-depth interviews. The analysis was performed by all teams of authors collectively in a scheduled group workshop for analysis. The team read through the transcribed data several times to gain overall meaning of the responses from the caregivers. After the team read through the transcribed data, the data were then arranged into subcategories and categories and labeled using the actual words and language of the caregivers. After developing and arranging data into various categories and subcategories, the team came up with themes which appeared to be the major findings of the study. In this case, four themes were generated.

Supervisors were independent coders who cross-checked the themes and categories to see if they were related to what the caregivers said during interviews, referring to the audio-recorder. The supervisors verified if the generated themes are related to the subcategories and a consensus was reached that the themes and subcategories are related. The authors then interpreted the findings of the study, wherein they explained what had been learnt with regard to caring for a child with cerebral palsy in the rural areas. The interpretation of findings is based on the researchers’ understanding about caring for children with cerebral palsy with the integration of various studies regarding caregivers of children with a cerebral palsy disability.

## 3. Results

The demographics of caregivers who were interviewed in this study are presented in Table 1 below. The table presents the relationship of the caregiver to the child, age of caregivers, caregivers employment status, and the child’s Gross Motor Function Classification System level.

Four main themes emerged from qualitative data collected through individual in-depth interviews with the participants, namely economic problems facing caregivers; health problems encountered by caregivers; maltreatment of caregivers by community members, as well as access to transport for caregivers and children.


**Theme 1: Economic problems facing caregivers**


The study revealed that caregivers who are caring for children with cerebral palsy experience economic challenges, due to the needs of the children and the costs of travelling to the health centers to access health services for the children.


**Child’s needs**


A majority of caregivers indicated that children with cerebral palsy have many needs. They reported that the children do not eat what other people in the house eat, and they have to use nappies for them because they cannot talk when they want to go to the toilet. The following quotes depict how the needs of children affect the caregivers economically:

“*…I have to buy him food that is different from what other children eat, he can’t chew anything. He wants soft food like potatoes, butternut and soft-porridge…*”

“*…I can buy soup and say he will eat and he doesn’t eat, am forced to go and buy other things. I will buy food and they will be many, when I feed him, he will be refusing, sa mukapu wa Ace, when I feed him he refuses, sometimes I buy one favourate (sic) and he doesn’t eat he refuses…*”

Some of the younger caregivers indicated that the disability grant money that they receive on behalf of the child is not enough to meet the needs of the children. The whole family depends on the grant, since they cannot work because they have to look after the children:

“*…He is getting that grant of disability…From the money that I will be holding, because when you look at the money that I get, It is not used for the child alone, we have to eat, we have to, everything is looking at the money of this child…*”

“*…This child does not eat everything, I have to buy him different things from what we eat, and everything in this house depends on the grant money of this child…*”

“*…My child is now big, when I buy pampers I have to buy two packets and they don’t even finish the month, and you must know that I am not working…*”

“*…It is difficult for me to work, If I go to work who will take care of my child? Another person won’t understand him like I do…*”


**Travelling costs**


Some of the caregivers reported that they have to use transport when taking the children to the cerebral palsy support group or when they have to collect items from the health centers. They have to pay for themselves and the children. One caregiver further stated that she once returned four times from the clinic when she went to collect the child’s wheelchair. The following quotes depict how caregivers are affected by the travelling costs:

“*…When I take him to the hospital I can’t use the bus, even though it’s cheaper because the seats are next to each other and my child won’t be comfortable. I am forced to use a taxi even if it is expensive, so that my child can be comfortable…*”

“*…Someday I went to the clinic, going to take his chair, and I was told to go back, I might have been returned 4 times, telling me that they are renovating, and it was boring me because I was using money for transport and am not working…*”


**Theme 2: Health problems encountered by caregivers**


Caregivers who are caring for children with cerebral palsy encounter various health problems as they provide care for the children, ranging from their physical health to experiencing negative attitudes held by the health professionals.


**Physical health**


Caring for a child with cerebral palsy poses a threat to the health of the caregivers. All caregivers raised concerns of back pains, which are a result of lifting the children because they cannot move on their own. The following quotes depict how caring for children with cerebral palsy affects the physical health of caregivers:

“*…Our nearest clinic is in Madimbo, so when the child is sick, I have to put him on my back and go there, if I don’t get a lift I will have to walk for that distance with this child on my back…*”

“*…I have an operation, if I carry this child, I know that am going to sleep when I come back because he is having…I know that tomorrow there will be clouds even when it’s hot like this because of this operation…*”

“*…It pains me because if he is like this and not sleeping, it means I will carry him on my back when am doing everything, if it’s doing my washing, cleaning and sweeping, everything he will be on my back. There is a pain that I know, when it’s cloudy like today I will have a pain on my back, because am talking about a person who is going to turn 6 years. He will be hurting me, I will be feeling pains…*”

“*…I cannot leave this child alone in the house when am cleaning outside or doing other thing. I have to put him on my back even when am cleaning the house and he is heavy, imagine carrying a seven years old person on my back. I always feel pains on my back…*”

“*…This child cannot do anything on his own. I have to carry him always. My back is always painful because this child is now big and heavy…*”


**Attitude of health professionals**


Several caregivers raised concerns regarding the health professionals’ attitude when they take their children for treatment or rehabilitation. They indicated that they go to the health centers and come back without any assistance, and the health professionals do not check their children. The following quotes depict how caregivers are subject to the negative attitude of health professionals:

“*…I once took my child to one of the local clinics, a tshi kho tou gomela (growning), when I got there the nurses took my baby’s clothes off and pressed his stomach, then they told me that he is not sick and they didn’t give him medication. I came back and went to the main road to ask for a lift to take my child to Madimbo clinic, when I got there my child was given medication, I don’t take my child to that clinic anymore. There is no service…*”

“*…When I take this child to the clinic for exercise, that man doesn’t do anything. He will put the child on the table and he will just stand on the other side and call the child. If the child looks at him, you will hear him saying, don’t worry your child will be fine. I want doctors to check my child, because that one is not doing anything…*”

One caregiver who is looking after her grandchild indicated that the nurses did not know how to talk to people, that they were shouting at her, saying she doesn’t look after her grandchild.

“*…One day I went to that house (pointing the house with a finger), this child was sleep and my husband was plastering the house that side (pointing at the side), when I came back I found the nurses who work around the villages with other nurses from Musina, they started shouting at me telling me that I am not taking care of my grandchild, those nurses don’t know how to talk to people. I am older than them they must respect me…*”


**Theme 3: Maltreatment of caregivers by community members**


Caregivers of children with cerebral palsy in this study revealed that they are being maltreated by other community members in the villages where they are staying. They indicated that while some people always involve the child’s disability in their fights with the caregivers, others do not allow their children to play with the child with CP. As a result of that, caregivers isolate themselves from other people. The following quotes depict how caregivers are maltreated by other community members:

“*…In this community it is difficult, there are some people who insult me about my baby, because I already have two cases which were reported to the chief because they were talking about my child. I went to the police and they indicated that the people should ask for forgiveness. The first one agreed while we were still at the chief when she was been asked and accepted that she said it and she said she is asking for forgiveness. The second one refused and said it should be taken anywhere, it was then taken to the police station they talked to the person and she asked for forgiveness. It is not good if I don’t agree on something with the person and that person end up involving the child…*”

*“…When they see their children coming here, they start calling them. I think they don’t want their children to play with my child because he once bite another child’s finger…*”

“*…Here in the village we are not treated like other parents who gave birth like others, people talk somehow about us. If they say there are people who are registering to get things for the orphans, you will hear them saying a person who is getting the big grant is not supposed to get this. Someone said a person who has a child like this is not supposed to be given anything because the child is getting the big grant. The community here is not treating us right as people. They talk things, you even feel scared of walking around because when you come this side someone will be saying some things. It’s better to stay at home and look after the child, and only go out when am taking the child to the hospital…*”


**Theme 4: Access to transport for caregivers and children**


Caregivers need transport to move from their homes to the health facilities, shops, or other places. Certain caregivers raised concerns regarding access to proper transport in their areas. They indicated that they have to walk for a distance to access transport because the roads in the village are not good, taxis are always full when they reach their areas, they are always late for appointments, and they use bakkies because there are no taxis in their villages. The following quotes depict how caregivers face transport challenges in their areas:

“*…I am always late for the hospital, because there is no transport this side, sometimes my child does not get attended to because when I get there if find that the group is out…*”

“*…We don’t have taxis, we use bakkies when we want to go to town. And we have to wait for it to be full…*”

“*…This side we use taxis that are coming from Musina or Thohoyandou, and when they come here you will find that they are full of people. And am forced to stand inside the taxi with the child on my back. But if I find people with good hearts they give me a seat…*”

“*…Our roads are not good, and taxis don’t come here. I have to walk to the main road to get a taxi, carrying this child and it is a long distance from here…*”

## 4. Discussion

Caregivers of children with cerebral palsy from Vhembe District in South Africa encounter various challenges as a result of caring for a child with CP. The challenges that these caregivers encounter were found to be similar to what previous studies reported.

The findings of the study have revealed that both mothers and grandmothers who are caring for children with cerebral palsy face financial challenges, as a result of the high demands of children who do not eat the same food as what other members of the family are eating. [16] concurs with the findings that caring for children with cerebral palsy includes extra costs related to children’s trips to clinics and hospitals for treatment or rehabilitation, specific food that they need, and purchase of disposable nappies/diapers. Most caregivers in the study are not working since they are responsible for providing care to the children with cerebral palsy fulltime, making them to survive on the disability grant offered by the government. This study’s findings revealed that majority of young mothers also complained about the grant money not being enough to care for a child with cerebral palsy. This affirms the findings of [17] despite the government supporting children with a disability through disability grants. It is regarded as too little, since in most families the money has to cater for the needs of all people in the household, and only a little amount is directed towards the needs of the child with CP. Children who are suffering from cerebral palsy need continuous medical check-ups, since these children could be suffering from multiple conditions at the same time and that poses a threat to their health. Accessing the services required by these children was seen to be an overburden to caregivers due to limited financial budgets [18,19].

The findings of the study revealed that caregivers do not always receive the assistance they need at the health centers. All grandmothers complained about the way health professionals treat them. Both mothers and grandmothers experience backpains as a result of lifting the children with CP every day. These findings are consistent with those of scholars [20] who found that caregivers have difficulties in accessing services and often complain about the quality of the services that they receive from the public institutions. Ref. [21] also revealed that caregivers take up the role of an activist at health centers for their children to receive the services that they need. Ref. [8] states that caregivers feel that the services which are being offered at the health centers do not address the health challenges that they face, together with the children that they are caring for.

According to [3], trips to the hospitals are viewed as costly, not because of the amount that the caregivers pay for the transport, but due to the long queues at the hospitals as a result of not having adequate numbers of therapists or being assisted by therapists who are not experienced in dealing with CP cases. Parenthood of a disabled child is a proven risk factor for back problems in both male and female [8]. Ref. [8] further states that the major factor that is perceived to be contributing to back pain amongst the caregivers of children with a disability is the continuous lifting of heavy children, which affects their muscles and results in pains.

The study findings revealed that caregivers are not being treated well by other community members. Because of the condition of the child, they are excluded from receiving services and as a result they isolate themselves. Caregivers do not get support from family, friends, and community members. This results in them being socially isolated, which is partly as a result of fear amongst caregivers that people would not accept their children or blame them for the condition of the children [2].

Similarly, [10] found that the negative attitudes experienced from the community as a result of tradition and cultural beliefs with regard to disability has led to discrimination and prejudice against caregivers and the children with cerebral palsy. Ref. [2] also found that the response from the society causes caregivers of children with cerebral palsy to isolate themselves and keep their children at home.

The study revealed that accessing transport is a challenge to caregivers and the children with cerebral palsy in the rural areas, due to the condition of the roads and the remoteness of the villages. This concurs with the study done by [22] public transport was not well designed to accommodate people who use wheelchairs. A lack of well-equipped taxis and buses in the developed and developing countries force caregivers to hire private transport or use their own cars to transport their children. A lack of proper transport can result in caregivers and children with cerebral palsy withdrawing from public engagements [23].

The findings of the study cannot be generalized to the larger population of caregivers for children with cerebral palsy since some of the challenges might be influenced by cultural practices and geographical areas. The study was on the challenges encountered by caregivers. This study, therefore, recommends that an investigation be performed on what kind of model should be developed to support caregivers in caring for children with cerebral palsy.

## 5. Conclusions

Caring for children with cerebral palsy is a challenging and demanding task. Children who are being cared for have needs which the caregivers have to meet on a daily basis, and money is needed to meet some of those needs. As a result of providing care to the children with cerebral palsy, caregivers experience financial challenges, back pains, and they are sidelined in the communities where they live. Access to transport is a challenge to the caregivers and children with cerebral palsy.

## Figures and Tables

**Table 1 children-10-00440-t001:** Demographic data of the caregivers.

Participants No	Relationship to the Child	Age of Caregiver	Employment Status	Child’s Gross Motor Function Classification System
Participant 1	Mother	26	Unemployed	Level V
Participant 2	Mother	32	Unemployed	Level V
Participant 3	Mother	37	Unemployed	Level V
Participant 4	Mother	31	Unemployed	Level V
Participant 5	Mother	39	Self-Employed	Level V
Participant 6	Grandmother	57	Unemployed	Level V
Participant 7	Mother	31	Unemployed	Level V
Participant 8	Grandmother	49	Unemployed	Level V
Participant 9	Mother	28	Unemployed	Level V
Participant 10	Mother	34	Unemployed	Level V

## Data Availability

The anonymized data are available from the corresponding author upon request.

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
