# Peer review of "Title: Caring for Children with Cerebral Palsy: A Challenge to Caregivers in Rural Areas of South Africa"

_children, 2023, doi:10.3390/children10030440_

Round 1

Reviewer 1 Report

I am reviewing the article “Caring for children with cerebral palsy: A challenge to caregivers in rural areas of South Africa”. The article provided to present the challenges encountered by caregivers of children with cerebral palsy in rural areas of South Africa.  The sample number is few, and the paper had some problem that is misleading the results. Here are some comments on the article.

1. Both developed and developing country aspects of the issue need to be addressed in the introduction.

2. Information on carers of children with cerebral palsy participating in this study is scarce and inclusion and exclusion criteria are unclear, so additional information is needed.

3. The GMFCS and GMFM grades for children with cerebral palsy are unknown, and it is unclear what level of severity is covered. Please clarify information.

4. How did the authors determine the sample appropriate size?

5. If such a study is to be carried out, a very large sample is required. The sample size is insufficient to explain the results of this study.

6. The way in which the data from this study is presented is inadequate. More results with statistical analysis should be presented.

7. The paper itself is very interesting, but the overall impression is cumbersome. I think it is important to clarify the sample size and increase the sample size first.

Author Response

Thank you so much for the comments made. It really assisted us to improve the manuscript

Reviewer 2 Report

In the introduction, it is necessary to add a consideration of original studies on the burden of care for children with cerebral palsy. Please describe the contents of research on the burden of caring for children with cerebral palsy who are not children with disabilities.

The need to be carried out as a qualitative study should be mentioned in the introduction so that the rational reason for conducting this study can be understood.

In addition, it is necessary to add the importance and excellence of the results of qualitative research.

Although it was said that purposeful sampling was conducted by the researcher, there must be a valid explanation for the fact that bias did not exist in this purposeful sampling.

Since only the central question during interview was described, what are examples of additional questions that were added to the core question?

The main theme is divided into four, and detailed information on the process of deriving these main themes needs to be described.

And how did you confirm that this main theme was a valid result?

It would be good to present the practical meaning of the results of this study, such as the content that should be included in the development of a program that can help in caring for children with cerebral palsy.

Author Response

Thanks for the time you allocated to review our manuscript. Your comments were considered to improve our manuscript. 

Round 2

Reviewer 1 Report

I still have concerns regarding the sample size.

It is unlikely that the results of this very small data study will become general data for South Africa.

I would recommend authors to conduct a larger study before any conclusion can be made for these experiments.

Author Response

Thanks for the concerns and comments about the sample size. However, the method used in this manuscript is qualitative and the sample size is determined by data saturation. Hence, generalization applies in quantitative study.

Reviewer 2 Report

Thank you for your revisions.

Author Response

Thanks for the feedback